# Introducing Background Temperature to Characterise Hidden Randomness in Large Language Models

**Alberto Messina**[*]                                                             *alberto.messina@rai.it*
*RAI - Radiotelevisione Italiana, Centre for Research, Technological Innovation and Experimentation (CRITS)*

**Stefano Scotta**[*]                                                               *stefano.scotta@rai.it*
*RAI - Radiotelevisione Italiana, Centre for Research, Technological Innovation and Experimentation (CRITS)*

**Reviewed on OpenReview:** *https://openreview.net/forum?id=bz0he4bARF*

## Abstract

Even when decoding with temperature $T = 0$, large language models (LLMs) can produce divergent outputs for identical inputs. Recent works align in highlighting implementation-level sources of nondeterminism, including batch-size variation, kernel non-invariance, and floating-point non-associativity. In this work, we formalize this behavior by introducing the notion of *background temperature* $T_{\mathrm{bg}}$, the effective temperature induced by an implementation-dependent perturbation process observed even when nominal $T = 0$. We provide clean definitions, show how $T_{\mathrm{bg}}$ relates to a stochastic perturbation governed by the inference environment $I$, and propose an empirical protocol to estimate $T_{bg}$ via the equivalent temperature $T_n(I)$ of an ideal reference system. We conclude with a set of pilot experiments run on a representative pool from the major LLM providers that demonstrate the idea and outline implications for reproducibility, evaluation, and deployment.

## 1 Introduction

A common assumption in LLM deployment is that setting the decoding temperature to $T = 0$ (greedy decoding) ensures determinism. However, empirical evidence shows output variability persists under nominally deterministic settings. The recent work in He & Lab (2025) argues that nondeterminism in LLM inference often arises from practical systems issues such as varying batch sizes and the lack of batch-invariant kernels, along with floating-point non-associativity and reduction-order effects. Despite growing awareness of this phenomenon, current approaches to quantifying nondeterminism rely on isolated and often incompatible metrics—exact-match rates, edit distances, entropy changes, or perplexity-based measures (see Terven et al. (2025) and Yun et al. (2025)). While these capture specific facets of instability, they provide a fragmented picture and are of difficult usage to compare among models, providers, or hardware environments. This lack of an integrated approach prevents practitioners from reliably characterizing, monitoring and benchmarking the effective randomness of deployed systems.

To address this, we introduce the notion of a background temperature ($T_{\mathrm{bg}}$). This quantity summarizes the effective stochasticity induced by the models' inference environment, even when the configured decoding temperature is zero. By how it is defined, $T_{\mathrm{bg}}$ abstracts the specific stochastic sampling implementations of the various models, which affect each model's logits before decoding, instead providing the temperature of an idealized and shared reference system whose output-variability statistics best match those observed under real deployment conditions. This approach offers a unified, interpretable scalar that aggregates all sources of implementation noise into a consistent scale. Instead of tracking a bundle of heterogeneous metrics, practitioners obtain a single descriptor of inference instability, comparable across models, providers, and hardware.

---

[*]These authors contributed equally to this work.

The paper is organized by a first presentation of an heuristic definition and estimation protocol of background temperature, followed by a rigorous mathematical formalization. Finally, we validate the approach through extensive experiments.

**Contributions.** (i) A concise formal model that addresses the phenomenon of nondeterminism as a stochastic effect on the output probability; (ii) a formal definition of background temperature; (iii) the outline of a practical protocol to estimate it; (iv) a set of pilot experiments illustrating and validating the concept.

## 2 Related Work

### 2.1 Sources of non-determinism

LLM outputs can vary for several reasons. First, stochastic decoding methods (e.g., temperature, top-k, top-p) intentionally introduce randomness via sampling. Second, floating-point non-associativity makes large reductions (dot products, softmax, layer norms) sensitive to kernel implementations, parallelism patterns, and hardware, so small numerical differences can be amplified through the network and flip argmax decisions. Third, non–batch-invariant kernels combined with dynamic batching cause the numerics for a given request to depend on batch size and position within the batch, so changing server load leads to different outputs for the "same" prompt. Finally, system-level factors—such as parallelism configuration, non-deterministic library algorithms, and variation across training runs or checkpoints—introduce additional run-to-run variability even when the model architecture and code are nominally unchanged. The recent work by Thinking Machines Lab He & Lab (2025) provides a systems-first analysis of LLM nondeterminism, emphasizing batch-size variation and batch-invariant kernels for inference; they also explain how floating-point non-associativity and reduction ordering contribute to variability. A key, and sometimes underappreciated, driver of this variability is the effective batch size seen by the serving stack. Modern inference servers dynamically batch concurrent requests for throughput, and many GPU kernels used in the transformer forward pass—particularly matrix multiplications, attention, and normalization layers—are not batch-invariant: their floating-point reduction order changes as a function of batch size and, in some cases, of the position of a sequence within the batch. Because floating-point arithmetic is non-associative, different reduction orders induce small but systematic numerical differences, which can then be amplified by subsequent layers and eventually flip argmax decisions in the decoder, producing different token sequences for the same logical query. In this view, the inference engine is "run-to-run deterministic" given a fixed batch configuration, but from the perspective of an individual user facing time-varying server load, the stochasticity in batch composition translates directly into apparent nondeterminism in the model's outputs.

In addition to this work, similarly, several recent studies have quantified non-determinism in large language model outputs even under settings intended to be deterministic (e.g. temperature $T = 0$, fixed seeds). For example:

- Atil et al. (2025) systematically evaluate multiple LLMs configured under deterministic settings across zero-shot and few-shot tasks. They observe large accuracy variations (up to 15%) across runs with the same input, and show that even the string outputs are often not identical.

- Song et al. (2024) explore how evaluation practices often ignore variability arising from different decoding configurations (greedy vs sampling). They show that even for greedy decoding, evaluation metrics vary, and that alignment methods can help reduce sampling variance.

- Ouyang et al. (2025) analyze code generation benchmarks and show that many coding tasks produce different code outputs across repeated prompt invocations, even when using $T = 0$. This confirms that deterministic temperature settings do not guarantee output consistency.

- Yuan et al. (2025) show that changing system-level factors (GPU type, batch size, numeric precision) can alter LLM outputs even under greedy decoding.

## 2.2  Batch-invariant kernels

Batch-invariant kernels are numerical primitives whose outputs for a given logical example do not depend on the size of the enclosing batch, the example's position within that batch, or the way batches are assembled at serving time. Concretely, a batch-invariant matrix multiplication, RMSNorm, or attention kernel guarantees that running a single example in isolation, or as part of any larger batch, yields bitwise identical activations for that example, assuming fixed hardware and software versions. Achieving this property requires carefully constraining parallelization and reduction strategies: instead of allowing the compiler or kernel library to switch between different reduction trees (e.g., data-parallel vs. Split-K / stream-K) depending on batch size, batch-invariant kernels enforce a fixed, example-local reduction order that is reused across all batch shapes. This often trades some peak throughput for predictable numerics, but in practice the overhead can be modest when kernels are engineered to maintain sufficient parallelism without changing reduction topology. Recent LLM serving systems have begun to expose explicit "deterministic" or "batch-invariant" modes, which swap in such kernels for the critical operations (RMSNorm, matmul, attention) to decouple user-visible outputs from dynamic batching and to enable reproducible inference at scale.

While prior work largely documents the existence and magnitude of non-determinism, there remains a gap in formalizing this behavior in terms of an equivalent temperature transformation functional and in proposing standard protocols to measure the effective background randomness. Our work addresses this by introducing the notion of an equivalent temperature $T_n(I)$ and its expectation $T_{\mathrm{bg}}$. In the next sections, we transition from formal definitions to a concrete empirical protocol aimed at estimating an *equivalent temperature $T_n(I)$* induced by implementation noise, and ultimately the background temperature.

## 3  Heuristic Definition and Empirical Estimation

This section introduces $T_{bg}$ heuristically and outlines its estimation protocol. We start by defining the role of temperature in LLMs to establish the intuition behind $T_{\mathrm{bg}}$, followed by a concrete measurement procedure. A rigorous formalization is deferred to Section 4.

### 3.1  Preliminaries and Heuristic Definition

Let $D$ denote the vocabulary of a LLM, with cardinality $|D|$. For any token sequence $\tau \in D^{|\tau|}$, we write $\tau^t$ for its $t$-th token, and $\tau^{<t} = \{\tau^j : j < t\}$ for the subsequence of previously generated tokens. At each generation step $t \in \{1, \ldots, |\tau|\}$, the model produces a vector of logits $z_t = \{z_{t,\theta} : \theta \in D\} \in \mathbb{R}^{|D|}$ depending on the context $\tau^{<t}$. These logits are real-valued scores, one for each token $\theta \in D$. By applying the softmax function to $z_t$, we obtain a probability distribution over the vocabulary:

$$P_t(\theta) = P_t(\theta \mid \tau^{<t}) = \frac{\exp(z_{t,\theta})}{\sum_{\eta \in D} \exp(z_{t,\eta})}, \qquad \text{for all } \theta \in D. \tag{1}$$

A temperature parameter $T \geq 0$ is typically introduced to control the concentration of this probability distribution. The temperature acts by modifying the logits prior to the softmax by a scaling factor $1/T$. The resulting temperature-adjusted distribution is

$$P_t^T(\theta) = P_t^T(\theta \mid \tau^{<t}) = \frac{\exp(z_{t,\theta}/T)}{\sum_{\eta \in D} \exp(z_{t,\eta}/T)} \qquad \text{for all } \theta \in D. \tag{2}$$

For $T = 0$, the distribution concentrates its mass on the most likely token, effectively becoming a degenerate distribution (a Dirac delta) over $\arg\max_{\theta \in D} P_t(\theta)$. Conversely, as $T \to \infty$, the distribution flattens and converges to the uniform distribution $P_t^T(\theta) \to 1/|D|$. See also Li et al. (2025) for details.

It is convenient to interpret temperature as a transformation applied to the original distribution. We therefore define a temperature functional

$$F_T : \mathbb{R}^{|D|} \to \mathbb{R}^{|D|} \tag{3}$$

such that, given the base probability vector $P_t = P_t(\cdot) \in [0,1]^{|D|}$, one obtains the adjusted distribution $P_t^T = P_t^T(\cdot)$ by $P_t^T = F_T(P_t)$. In the idealized limit, $F_0$ corresponds to greedy decoding, assigning unit mass

to the most probable token, while $F_T$ with $T > 0$ produces a probability distribution used for sampling the next token in language models. In real LLM deployments, the effective distribution obtained at temperature $T = 0$ may exhibit nondeterministic behavior due to implementation-dependent factors. As observed by the authors in He & Lab (2025), even nominally deterministic decoding ($T = 0$) can produce variability in the generated token sequence. Let $I$ denote the inference environment, which includes hardware characteristics, backend implementations, numerical precision, batch composition, concurrency, reduction ordering, and other low-level sources of variation (see He & Lab (2025) and Shanmugavelu et al. (2024), for more details). Let $F_T'$ denote the temperature transformation functional realized by the system under environment $I$. We model these effects as a perturbation of the ideal functional $F_T$. Specifically, we assume that there exists a perturbation mapping $\epsilon_I$ from probability distributions over $D$ to probability distributions over $D$ such that, at $T = 0$,

$$F_0'(P_t) = \epsilon_I(F_0(P_t)) = \epsilon_I(P_t). \tag{4}$$

Even if the changes given by $\epsilon_I$ are small in magnitude, regions of the probability simplex where two or more tokens have similar mass are highly sensitive: arbitrarily small changes may alter the $\arg\max$ and therefore change the emitted token at step $t$.

We interpret the effect of $\epsilon_I$ as producing a distribution that behaves approximately as if sampling were performed at a nonzero temperature, even when $T = 0$ has been selected. We therefore postulate the existence of an equivalent temperature $T_n(I)$ such that

$$F_0'(P_t) = \epsilon_I(P_t) = F_{T_n(I)}(P_t). \tag{5}$$

The background temperature of an LLM implementation is the expected equivalent temperature over all the possible inference environments $\mathcal{I}$, so, heuristically, it is given by

$$T_{\mathrm{bg}} := \mathbb{E}_{I \in \mathcal{I}}\left[T_n(I)\right]. \tag{6}$$

Intuitively, $T_{\mathrm{bg}}$ quantifies the implicit randomness present in real systems even when the user explicitly requests deterministic next-token selection. The quantity $T_{\mathrm{bg}}$ thus captures implementation-level nondeterminism and paves the way for an effective measure of intrinsic variability in practical LLM decoding pipelines.

### 3.2 Estimating $T_n(I)$ and $T_{\mathrm{bg}}$ Empirically

The heuristic definition in equation 6 involves an abstract expectation over inference environments, which cannot be observed directly. To make $T_n(I)$ and $T_{\mathrm{bg}}$ empirically measurable, we replace the ideal reference system with a quasi-ideal baseline: a controlled inference setup in which known sources of nondeterminism - batch-dependent kernels, floating-point precision issues, concurrency, reduction order, and framework-specific flag- are minimized. Prior works shows that such configurations significantly reduce output variability under zero temperature, making them suitable as reference systems. Running the models locally guarantees a stable inference environment.

The empirical estimation protocol proceeds as follows. First, one selects a diverse and representative prompt set $\Pi$, including general generation prompts (short or long, with common or rare vocabulary), task benchmarks such as QA (e.g., truthful_qa Lin et al. (2022), SQuADPrice & Cote (2025), TriviaQAJoshi et al. (2017)), summarization, classification, or code generation, edge or adversarial prompts (e.g., long contexts, rare tokens, top-$k$ ties), and so on.

Second, repeated inference is run at $T = 0$ with the system under test, while systematically varying the inference environment $I$ along axes known to influence nondeterminism. These include batch structure and co-batching, concurrency and load, hardware/backend types (GPU/CPU, numeric precision, kernel implementations), and numerics such as reduction order, deterministic flags, or fused versus unfused kernels (see He & Lab (2025), Shanmugavelu et al. (2024), Ravi et al. (2025) for details). For remote systems, prolonged and repeated operation can be used to statistically sample the distribution of inference environments.

Third, under the stable baseline environment, the same prompts are run across a grid of temperatures to establish a mapping between temperatures and output-variability statistics.

Fourth, variability metrics are computed across multiple runs to capture the system's nondeterminism. Metrics can include exact-match rates, first-divergence indices, edit distances, distributional divergences such as KL or JS divergence, and entropy of next-token distributions.

Fifth, the distributions of variability metrics are compared between the system under test and the reference system. The temperature of the reference system that minimizes some chosen distance between distributions is taken as the equivalent temperature for that environment. Repeating this across prompts and environmental samples yields an empirical estimate of $T_n(I)$.

Finally, aggregating the equivalent temperatures across sampled environments and prompts produces an estimate of the system's background temperature $T_{\text{bg}}$, capturing the intrinsic nondeterminism in real-world LLM deployments. Uncertainty bounds can be derived from the variability across prompts, environments, and reference models. This protocol allows measurement of implicit randomness in LLMs without requiring access to a perfect oracle, while providing guidance for system design and operational mitigation of nondeterminism.

## 4 Mathematical Formalization

### 4.1 Rigorous Definitions

In this Section we provide a rigorous definition of $T_{\text{bg}}$, generalizing it to the case in which the temperature of system under test is not set to 0 but to a generic $\tau > 0$. First of all we introduce all the necessary notation that we are going to use.

#### 4.1.1 Notation

We denote by $\bar{\mathcal{L}}$ the set of all possible LLMs $\ell$. Let $\mathcal{S}$ denote the set of pairs $s = (m, \alpha)$, where $m \in \bar{\mathcal{L}}$ and $\alpha$ is the provider through which $m$ is accessed. Note that if there is no provider — meaning that the LLM is run locally — we simply denote $s = (m, 0) = m$. Closely related to the models accessed through a provider is the set $\bar{\mathcal{I}}$ of all possible inference environments in which a system $s \in \mathcal{S}$ can be executed. The temperatures used in the LLM configurations are denoted by $T \in [0, T_{\max}]$, where $T_{\max}$ is a finite arbitrarily large value in $\mathbb{R}_+$. We denote by $\bar{\Pi}$ the set of all possible prompts.

Then we introduce the concept of variability measures, which are functions that take as input a set of $N$ responses produced by an LLM and return a scalar (or vector) value that quantifies the variability among them. The set of all variability measures is denoted by $\bar{\mathcal{V}}$. The empirical distribution of the values of $v \in \bar{\mathcal{V}}$ computed for each prompt in $\bar{\Pi}$ over $N$ answers generated by $\ell \in \bar{\mathcal{L}}$ at temperature $T \in [0, T_{\max}]$ is denoted by $f_T^\ell(v) = f_T^\ell(v, \bar{\Pi})$. Analogously, the empirical distribution of the values of $v \in \bar{\mathcal{V}}$ computed for each prompt in $\bar{\Pi}$ over the $N$ answers generated by $s \in \mathcal{S}$ at temperature $\tau \in [0, T_{\max}]$ in the inference environment $I \in \bar{\mathcal{I}}$ is denoted by $g_\tau^s(v, I) = g_\tau^s(v, \bar{\Pi}, I)$. Note that both $f_T^\ell(v)$ and $g_\tau^s(v)$ are elements of $cod(v)^{|\bar{\Pi}|}$, where $cod(v)$ denotes the codomain of $v$. Lastly, for each $v \in \bar{\mathcal{V}}$, we denote by $d_v$ a function that evaluates the distance (or similarity) between variability distributions computed with $v$. For example, if $v$ has codomain $\mathbb{R}^N$, $d_v$ could be the Kolmogorov-Smirnov distance (see Wilcox (2010)), which quantifies the distance between two distributions.

#### 4.1.2 Definitions

Hereinafter, let us assume that all the sets $\bar{\Pi}, \bar{\mathcal{V}}, \bar{\mathcal{I}}, \bar{\mathcal{L}}$ and $\mathcal{S}$ are finite and discrete. This could seem restrictive in general, but in practice, it is trivially true. First, we define precisely the equivalent temperature $T_n$ introduced in Section 3.1.

**Definition 4.1** *Consider the probability space $(\Omega, \mathcal{F}, p)$, where $\Omega = \bar{\mathcal{I}} \times \bar{\mathcal{V}} \times \bar{\mathcal{L}}$, $\mathcal{F} = 2^\Omega$, and $p$ is the uniform probability measure on $\Omega$. For any $\ell \in \bar{\mathcal{L}}$, $s \in \mathcal{S}$, $v \in \bar{\mathcal{V}}$, $I \in \bar{\mathcal{I}}$, $\tau \in [0, T_{\max}]$, and a chosen distance function*

$d_v$, we define the equivalent temperature $T_{s,\tau}^\ell(I, v)$ as the value of the random variable

$$
\begin{aligned}
T_{s,\tau} : \Omega &\longrightarrow [0, T_{\max}], \\
(I, v, \ell) &\longmapsto T_{s,\tau}^\ell(I, v) := \frac{1}{|\mathcal{M}_{s,\tau}^\ell(I, v)|} \sum_{t \in \mathcal{M}_{s,\tau}^\ell(I, v)} t,
\end{aligned}
\tag{7}
$$

where

$$
\mathcal{M}_{s,\tau}^\ell(I, v) := \operatorname*{argmin}_{T \in [0, T_{\max}]} d_v\big(f_T^\ell(v), g_\tau^s(v, I)\big)
\tag{8}
$$

and the set of minimizers $\mathcal{M}_{s,\tau}^\ell(I, v)$ is assumed to be nonempty (which holds, for instance, if $T \mapsto d_v(f_T^\ell(v), g_\tau^s(v, I))$ is continuous on the compact interval $[0, T_{\max}]$).

In words, $T_{s,\tau}^\ell(I, v)$ is defined as the mean of all temperatures in $[0, T_{\max}]$ that minimize the distance between the variability distributions $f_T^\ell(v, \Pi)$ and $g_\tau^s(v, \Pi, I)$. Given the assumptions it is easy to prove that $T_{s,\tau}$ is a measurable random variable. Now, we can finally define rigorously the background temperature of a system.

**Definition 4.2** *The background temperature $T_{\mathrm{bg}}^\tau(s)$ of a system $s \in \mathcal{S}$ run at temperature $\tau \in [0, T_{\max}]$ is the value of the function*

$$
\begin{aligned}
T_{\mathrm{bg}} : \mathcal{S} \times [0, T_{\max}] &\longrightarrow [0, T_{\max}] \\
(s, \tau) &\longmapsto T_{\mathrm{bg}}^\tau(s) := \mathbb{E}_p[T_{s,\tau}^\ell(I, v)],
\end{aligned}
\tag{9}
$$

*at the point $(s, \tau)$. Where the expectation is taken with respect to the probability measure $p$ introduced in Definition 4.1.*

Definition 4.2 is quite general and takes into account the possibility of estimating background temperature for systems run at non 0 temperatures, this goes beyond the intuitive idea given in Section 3.1 but it generalize that concept and will be useful for next Sections.

### 4.2 Estimation Procedure

Clearly, when we want to estimate the background temperature for a system $s \in \mathcal{S}$, working with the sets used in Section 4.1 is practically impossible. To make estimation tractable, we restrict ourselves to finite subsets of the original domains: a reduced set of reference LLMs $\mathcal{L} \subset \bar{\mathcal{L}}$; a subset of prompts $\Pi \subset \bar{\Pi}$; a limited set $\mathcal{I} \subset \bar{\mathcal{I}}$ of inference environments; a finite discrete subset of temperatures $\Theta \subset [0, T_{\max}]$; a selection of variability measures $\mathcal{V} \subset \bar{\mathcal{V}}$. Thus, the distributions $f_T^\ell(v)$ and $g_\tau^s(v)$, have to be intended as $f_T^\ell(v, \Pi)$ and $g_\tau^s(v, \Pi, I)$, for $T, \tau \in \Theta$, $\ell \in \mathcal{L}$, $v \in \mathcal{V}$.

Then, adapting the definitions in Section 4.1, the estimated equivalent temperature is defined as

$$
\hat{T}_{s,\tau}^\ell(I, v) := \frac{1}{|\hat{\mathcal{M}}_{s,\tau}^\ell(I, v)|} \sum_{t \in \hat{\mathcal{M}}_{s,\tau}^\ell(I, v)} t, \quad \text{where} \quad \hat{\mathcal{M}}_{s,\tau}^\ell(I, v) := \operatorname*{argmin}_{T \in \Theta} d_v\big(f_T^\ell(v), g_\tau^s(v, I)\big).
\tag{10}
$$

That is, $\hat{T}_{s,\tau}^\ell(I, v)$ is the mean of all discrete temperatures in $\Theta$ that minimize the distance between the empirical variability distributions $f_T^\ell(v)$ and $g_\tau^s(v, I)$. Consequently, the estimated background temperature is given by the empirical mean

$$
\bar{T}_{\mathrm{bg}}^\tau(s) := \frac{1}{|\mathcal{I}| \, |\mathcal{L}| \, |\mathcal{V}|} \sum_{I \in \mathcal{I}} \sum_{\ell \in \mathcal{L}} \sum_{v \in \mathcal{V}} \hat{T}_{s,\tau}^\ell(I, v).
\tag{11}
$$

Clearly, the larger the sets $\mathcal{I}$, $\mathcal{L}$, and $\mathcal{V}$, the more robust the estimate $\bar{T}_{\mathrm{bg}}^\tau(s)$ will be.

The focus through this work will be on estimating the value of $\bar{T}_{\mathrm{bg}}^0(s)$ for different systems $s$, but we will also use the estimates $\bar{T}_{\mathrm{bg}}^\tau(s)$ for generic temperatures $\tau$ for validating the pipeline.

# 5 Experiments setting

The experiments were conducted using the following settings (see notation in Sections 4.1.1 and 4.2):

- $\Pi = \Pi^n$, consisting of the first $n$ prompts in TruthfulQA[1] (Lin et al. (2022)) dataset. Each prompt was used $N = 100$ and $N = 32$ times for each system under test and reference model, respectively, limiting the answers to 32 tokens.

- $\Theta = \{0, 0.01, \ldots, 0.19, 0.2, 0.25, \ldots, 0.45, 0.5, 0.6, \ldots, 0.9, 1\}$.

- The reference LLMs considered are $\mathcal{L} = \{\text{smoll}, \text{llama}, \text{mistral}\}$, where smoll, llama and mistral stand for, respectively, SmolLM3-3B[2] (Bakouch et al. (2025)), Llama-3.2-3B-Instruct[3] and Mistral-7B-Instruct-v0.3[4] (Jiang et al. (2023)).

- In most of the experiments, we considered a single inference environment $I$, corresponding to the conditions under which the experiment was run. Unless otherwise specified, the set $\mathcal{I}$ therefore consists of this single element $I$.

- The distance that we use between the distributions of the variability–metric values is always the Kolmogorov–Smirnov distance and we denote it by $d_{KS}$. This choice is appropriate because it quantifies discrepancies between one-dimensional distributions, and all variability metrics in our set $\mathcal{V}$ produce one-dimensional real values (see Section 5.1).

In the following Section, we define rigorously the variability metrics, i.e. the set $\mathcal{V}$, that we used. The set $\mathcal{S}$ of the systems under test will be specified for each experiment.

## 5.1 Variability Measures

We used the following variability measures to quantify the variability within the $N$ answers given by the LLMs to each prompt. See Figure 1 for a more in-depth analysis of them. Throughout this section let us denote for each prompt $\pi \in \Pi$, the generated answers by $A_\pi = \{a_1^\pi, \ldots, a_N^\pi\}$.

### 5.1.1 Maximum Exact Match Fraction

Let $\varphi_\pi$ be the frequency of the most common answer in $A_\pi$. The maximum exact match fraction for the answers given to $\pi \in \Pi$ is defined as

$$\text{MEMF}(\pi) := \frac{\varphi_\pi}{N}, \tag{12}$$

which measures how often the model produces exactly the same answer. $\text{MEMF}(\pi) = 1$ in the case in which all the answers to $\pi$ are identical, and it is $\text{MEMF}(\pi) = 1/N$ if all the answers are unique.

### 5.1.2 Average Normalized Levenshtein Distance

Let $\text{Lev}(a, b)$ denote the Levenshtein distance between two lists of tokens, i.e. the number of elements to change, add or remove from $a$ to make it identical to $b$. The average normalized Levenshtein distance for the $N$ answers given to $\pi \in \Pi$ is defined as

$$\text{AvgLev}(\pi) := \frac{1}{\binom{N}{2}} \sum_{1 \leq i < j \leq N} \frac{\text{Lev}(a_i^\pi, a_j^\pi)}{L_{max}}, \tag{13}$$

where $L_{max}$ is a normalization factor. Note that in our context $L_{max} = 32$, indeed, being the answers capped to 32 tokens, the maximum value of $\text{Lev}(a_i^\pi, a_j^\pi)$, for any $i, j \in \{1, \ldots, N\}$ and $\pi \in \Pi$, is 32. It goes from 0, when all answers are identical to 1 where they are all completely different.

---

[1] https://huggingface.co/datasets/truthfulqa/truthful_qa
[2] https://huggingface.co/HuggingFaceTB/SmolLM3-3B
[3] https://huggingface.co/meta-llama/Llama-3.2-3B-Instruct
[4] https://huggingface.co/mistralai/Mistral-7B-Instruct-v0.3

### 5.1.3 Average Longest Common Subsequence Distance

Let $\mathrm{LCS}(a, b)$ denote the length of the Longest Common Subsequence between two sequences of tokens $a$ and $b$. We define the corresponding normalized distance as

$$\mathrm{DistLCS}(a, b) := 1 - \frac{\mathrm{LCS}(a, b)}{\min(|a|, |b|)}. \tag{14}$$

Thus, $\mathrm{DistLCS}(a, b) = 0$ for identical answers and approaches 1 when the two answers share no common subsequence. For each prompt $\pi \in \Pi$, the average LCS distance is defined as

$$\mathrm{AvgLCS}(\pi) := \frac{1}{\binom{N}{2}} \sum_{1 \leq i < j \leq N} \mathrm{DistLCS}\left(a_i^\pi, a_j^\pi\right). \tag{15}$$

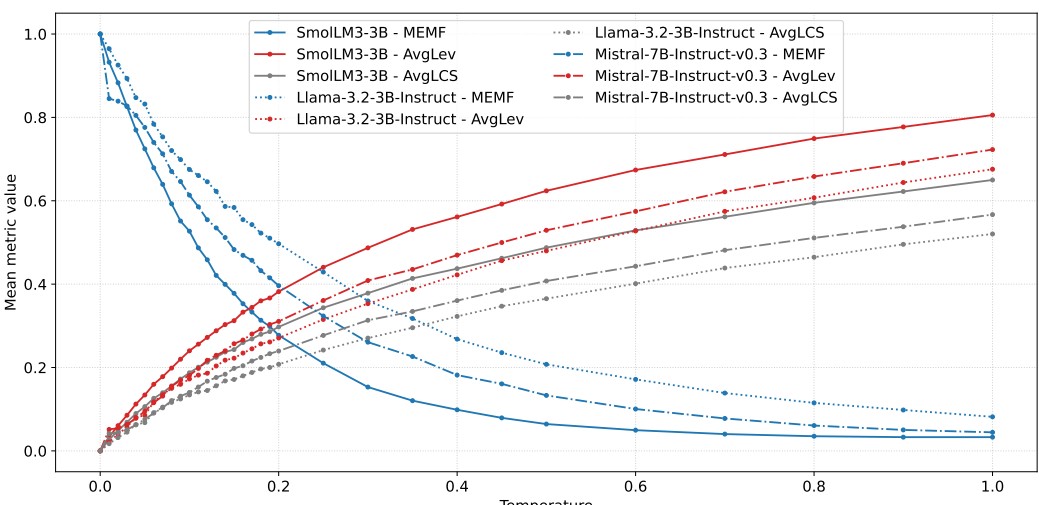

Figure 1: Mean values of the metrics defined below as a function of the temperature, computed over the answers produced by the reference models in $\mathcal{L}$ to the prompts in $\Pi^{30}$, each sampled 32 times.

## 6 Experiments and Results

In this section, we present some experiments to validate the theory presented in this work. The pipeline described in Section 4.2, under the experimental setting in Section 5, is implemented in `https://github.com/RaiCRITS/background-temperature-estimation`.

### 6.1 Background Temperature Estimate Across Models and Providers

In this experiment, we estimate the background temperature using the pipeline and the settings presented in Sections 4.2 and 5, for a set of different systems $s \in \mathcal{S}$ run at zero temperature. The goal is to determine whether different LLMs, accessed through different providers, exhibit significantly different variability in their answers, and whether the background temperature can capture this property. We used the same prompt set $\Pi^{30}$ for all the systems considered, and we also tested whether significant changes appear when increasing the prompt set in the case $s = (\text{gpt-4.1-nano}, \text{Azure})$, in which we also considered $\Pi^{200}$, to see if this change modifies significantly the estimates. All results are reported in Table 1.

Estimated background temperatures $\bar{T}_{\mathrm{bg}}^0(s)$ vary systematically across systems $s \in \mathcal{S}$, spanning an order of magnitude from 0 (claude-sonnet-4 on AWS) to 0.288 (DeepSeek-V3.1 on Azure). This pattern of variability indicates that the background temperature is not determined solely by the underlying model architecture, but instead emerges from the specific model–provider pair. The case of DeepSeek-V3.1 makes this especially

Table 1: Values of $\bar{T}_{\mathrm{bg}}^0(s)$ where $s$ is the combination of the first two columns. The column "Region" gives additional information about the inference environment considered.

| LLM | Provider | Region | $\bar{T}_{\mathrm{bg}}^0(s)$ | Prompt Set |
|---|---|---|---|---|
| claude-sonnet-4 | AWS | eu-west-3 | 0.000 | $\Pi^{30}$ |
| grok-3-mini | Azure | swedencentral | 0.016 | $\Pi^{30}$ |
| mistral-large-2402-v1:0 | AWS | us-east-1 | 0.016 | $\Pi^{30}$ |
| gemini-2.0-flash | Google | – | 0.048 | $\Pi^{30}$ |
| gpt-4.1-nano | Azure | eastus2 | 0.068 | $\Pi^{200}$ |
| gpt-4.1-nano | Azure | eastus2 | 0.087 | $\Pi^{30}$ |
| DeepSeek-V3-0324 | Azure | swedencentral | 0.148 | $\Pi^{30}$ |
| DeepSeek-V3.1 | AWS | us-west-2 | 0.186 | $\Pi^{30}$ |
| DeepSeek-V3.1 | Azure | swedencentral | 0.288 | $\Pi^{30}$ |

clear: the model deployed on AWS showed $\bar{T}_{\mathrm{bg}}^0(s) = 0.186$, whereas the same model on Azure increases to 0.288, despite the model and the experimental protocol being identical in both settings. Another interesting observation is that claude-sonnet-4 on AWS behaved perfectly deterministically over the tested prompt set: for each prompt, it produced the same sequence of tokens across all $N = 100$ iterations, resulting in a background temperature of 0. While this does not guarantee determinism for arbitrary prompt sets or token limits, it suggests that the configuration $s = (\text{claude-sonnet-4}, \text{AWS})$ represented the most stable system among those tested. Overall, these results indicate that $\bar{T}_{bg}^0(s)$ captures a consistent signature of variability that depends on the specific model–provider combination, rather than on the LLM model alone.

## 6.2 Validation of Background Temperature via Configured Sampling Temperature

To further validate the background temperature as a consistent measure of LLM response variability, we evaluated $\bar{T}_{\mathrm{bg}}^\tau(s)$ for multiple values of the configured sampling temperature $\tau$ across different systems and providers, keeping all other inference settings fixed. We also included an on-premises model, gemma-3-1b-it[5] (Team (2025)), providing a fully controlled environment that serves as a baseline for comparison against cloud-hosted systems. The tested systems and the obtained values are reported in Table 2.

Table 2: Values of $\bar{T}_{bg}^\tau(s)$ for different values of $\tau$ and systems $s \in \mathcal{S}$.

| LLM | Provider | Region | $\tau$ | $\bar{T}_{\mathrm{bg}}^\tau(s)$ | $\Pi^n$ |
|---|---|---|---|---|---|
| gpt-4.1-nano | Azure | eastus2 | 0.00 | 0.068 | $\Pi^{200}$ |
| gpt-4.1-nano | Azure | eastus2 | 0.05 | 0.081 | $\Pi^{200}$ |
| gpt-4.1-nano | Azure | eastus2 | 0.10 | 0.101 | $\Pi^{200}$ |
| grok-3-mini | Azure | swedencentral | 0.00 | 0.016 | $\Pi^{30}$ |
| grok-3-mini | Azure | swedencentral | 0.05 | 0.032 | $\Pi^{30}$ |
| grok-3-mini | Azure | swedencentral | 0.10 | 0.066 | $\Pi^{30}$ |
| claude-sonnet-4 | AWS | eu-west-3 | 0.00 | 0.000 | $\Pi^{30}$ |
| claude-sonnet-4 | AWS | eu-west-3 | 0.05 | 0.018 | $\Pi^{30}$ |
| claude-sonnet-4 | AWS | eu-west-3 | 0.10 | 0.022 | $\Pi^{30}$ |
| gemma-3-1b-it | On-premises | – | 0.00 | 0.000 | $\Pi^{200}$ |
| gemma-3-1b-it | On-premises | – | 0.05 | 0.032 | $\Pi^{200}$ |
| gemma-3-1b-it | On-premises | – | 0.10 | 0.061 | $\Pi^{200}$ |
| gemma-3-1b-it | On-premises | – | 0.15 | 0.093 | $\Pi^{200}$ |

The estimated background temperatures generally do not match exactly the configured sampling temperatures, even in fully controlled on-premises settings. However, $\bar{T}_{bg}^\tau(s)$ consistently increases with $\tau$ across all

---

[5]https://huggingface.co/google/gemma-3-1b-it

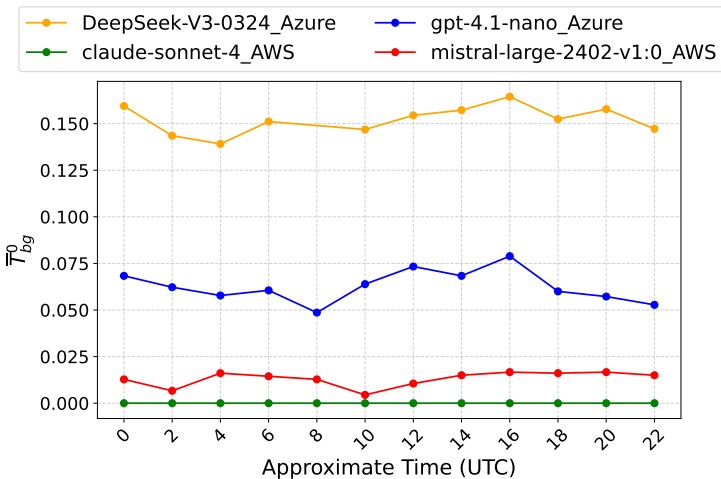

Figure 2: Evolution of $\bar{T}_{bg}^0(s)$ for different $s \in \mathcal{S}$ (see Table 1 for the regions and prompt sets) over the course of a day, highlighting the effect of temporal changes in the inference environment.

systems, demonstrating a monotonic relationship:

$$\tau_1 < \tau_2 \implies \bar{T}_{bg}^{\tau_1}(s) < \bar{T}_{bg}^{\tau_2}(s) \quad \forall s \in \mathcal{S}. \tag{16}$$

For example, claude-sonnet-4 on AWS consistently exhibits the lowest background temperature among cloud-hosted models, confirming the trends observed in the zero-temperature experiments. These results indicate that $\bar{T}_{bg}^{\tau}(s)$ provides a meaningful and interpretable estimate of the effective stochasticity in the model outputs. Importantly, it captures the variability of responses independently of the configured sampling temperature $\tau$, which can be influenced by provider-specific implementations and the inference environment. Therefore, $\bar{T}_{bg}^{\tau}(s)$ can be considered a more uniform and intrinsic measure of output variability, allowing for meaningful comparisons across models and deployment environments regardless of the temperature setting.

## 6.3 Temporal Variability of $\bar{T}_{bg}^0(s)$

To investigate the stability of background temperature estimates over time, hypothesizing that the time is a relevant component of the inference environment, we repeated the estimation procedure for a selection of LLMs from different providers at multiple times throughout the day (Figure 2). All models exhibited some temporal fluctuations in $\bar{T}_{bg}^0(s)$, except for claude-sonnet-4 accessed via AWS, which remained perfectly stable throughout the day, confirming the results discussed in Section 6.1. Despite these variations, the trajectories of different systems did not intersect, indicating that temporal fluctuations are smaller than those induced by other inference-environment factors. To clarify whether these temporal fluctuations were due to intrinsic variability of the models or to changes in the inference environment, we performed five concurrent estimations of $\bar{T}_{bg}^0(s)$ using `gpt-4.1-nano` on Microsoft Azure, keeping all conditions constant (same provider, model, execution time) and using the prompt set $\Pi^{30}$. The fluctuations observed across these concurrent runs were

Table 3: $\bar{T}_{bg}^0$ statistics for gpt-4.1-nano via Azure (swedencentral). "Same-time" refers to five contemporary estimates, "Day-long" refers to the experiment in Section 6.3 and Figure 2.

| Experiment | Mean $m$ | Std. Dev. $\sigma$ | $\sigma/m$ |
|---|---|---|---|
| Same-time | 0.0985 | 0.0038 | 0.039 |
| Day-long | 0.0650 | 0.0085 | 0.131 |

minimal (Table 3) and considerably smaller than those observed in the non-simultaneous measurements. This demonstrates that the temporal variations seen in the first experiment are primarily driven by changes in the inference environment over time, rather than by intrinsic randomness of the models under identical

conditions. In general, these results show that temporal aspects are relevant components of the inference environment, significantly changing the estimates $\bar{T}_{bg}^{0}(s)$, i.e. influencing the stability of the answers given by the model.

## 6.4 Validation of $\bar{T}_{\mathrm{bg}}^{\tau}(s)$ as a Measure of Answer Variability

To further assess whether the estimated background temperature $\bar{T}_{\mathrm{bg}}^{\tau}(s)$ provides a reliable measure of the variability exhibited by a system $s \in \mathcal{S}$ at a given sampling temperature $\tau$, we evaluated the relationship between $\bar{T}_{bg}^{\tau}(s)$ and the empirical variability of model outputs on a new dataset. For this purpose, we considered a prompt set $\tilde{\Pi}$ consisting of the first 100 instructions from the test split of the Hugging-FaceH4/testing_self_instruct_small[6] dataset. We selected three systems whose estimated background temperatures (see Table 2) allowed for a controlled comparison: gpt-4.1-nano deployed on Azure with $\tau = 0$, gemma-3-1b-it running on-premises with $\tau = 0.10$, which share similar values of $\bar{T}_{\mathrm{bg}}^{\tau}(s)$ (respectively, 0.068 and 0.061); and gemma-3-1b-it on-premises with $\tau = 0.15$, whose estimated background temperature (0.093) is noticeably higher. This setup enables testing the hypothesis that systems with similar $\bar{T}_{bg}^{\tau}(s)$ exhibit similar distributions of output variability on a given prompt set (different from the one used to estimate $\bar{T}_{bg}^{\tau}(s)$), independently of model architecture and deployment environment. Thus, we analyzed the distributions of response variability (computed with the measures in Section 5.1) over the prompt set $\tilde{\Pi}$, using each prompt 100 times. Pairwise differences between the variability distributions were quantified using the Kolmogorov–Smirnov distance $d_{KS}$ (Figure 3) across all variability metrics $v \in \mathcal{V}$. Additionally, the Wilcoxon

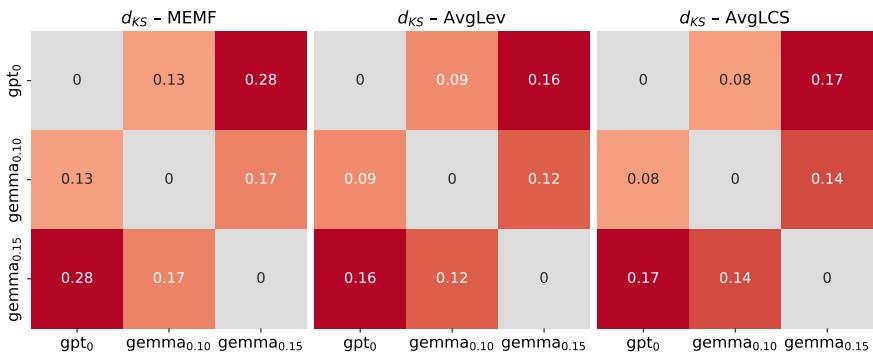

Figure 3: $d_{KS}$ between the variability distributions on the answers given to the prompts in $\tilde{\Pi}$, by the models gpt-4.1-nano, via Azure at temperature 0, denoted by $\mathrm{gpt}_0$, and gemma-3-1b-it, on-premises at temperatures 0.10 and 0.15, denoted respectively by $\mathrm{gemma}_{0.10}$ and $\mathrm{gemma}_{0.15}$.

signed-rank test (see Wilcox (2010)) was applied to assess whether the distributions corresponding to the two models with similar $\bar{T}_{bg}^{\tau}(s)$ can be considered statistically analogous, while the third model is expected to exhibit a different distribution. The results, summarized in Figure 3 and Table 4, clearly indicate that models with similar background temperatures produce variability distributions that are significantly closer to each other than to those of the higher-temperature system. Across all metrics, the Wilcoxon test fails to reject the null hypothesis of analogous distributions for the pair with matching $\bar{T}_{bg}^{\tau}(s)$, whereas comparisons involving the third model consistently reject it.

Overall, these findings confirm that $\bar{T}_{bg}^{\tau}(s)$ effectively captures the intrinsic variability of responses induced by a given sampling temperature. Moreover, the consistency of the results across datasets and metrics demonstrates that background temperature is a robust and reliable proxy for measuring answer variability in LLMs.

---

[6]https://huggingface.co/datasets/HuggingFaceH4/testing_self_instruct_small

Table 4: Wilcoxon signed-rank test results for variability comparisons between model pairs across for $v \in \mathcal{V}$. The model are denoted as follows: gpt-4.1-nano, via Azure at temperature 0, by $\text{gpt}_0$, and gemma-3-1b-it, on-premises at temperatures 0.10 and 0.15, denoted, respectively, by $\text{gemma}_{0.10}$ and $\text{gemma}_{0.15}$. The bold p-values are the cases in which we can not reject the null hypothesis $H_0$ that the distributions are analogous, in the other cases we reject $H_0$.

| Metric | Model Pair | Wilcoxon Stat | p-value |
|--------|-----------|---------------|---------|
| MEMF | $\text{gpt}_0$ vs $\text{gemma}_{0.10}$ | 1090.0 | **0.05342** |
| MEMF | $\text{gpt}_0$ vs $\text{gemma}_{0.15}$ | 727.5 | 0.00008 |
| MEMF | $\text{gemma}_{0.10}$ vs $\text{gemma}_{0.15}$ | 50.5 | 0.00000 |
| AvgLev | $\text{gpt}_0$ vs $\text{gemma}_{0.10}$ | 1398.0 | **0.73647** |
| AvgLev | $\text{gpt}_0$ vs $\text{gemma}_{0.15}$ | 1097.0 | 0.03999 |
| AvgLev | $\text{gemma}_{0.10}$ vs $\text{gemma}_{0.15}$ | 33.0 | 0.00000 |
| AvgLCS | $\text{gpt}_0$ vs $\text{gemma}_{0.10}$ | 1276.0 | **0.54805** |
| AvgLCS | $\text{gpt}_0$ vs $\text{gemma}_{0.15}$ | 939.0 | 0.01568 |
| AvgLCS | $\text{gemma}_{0.10}$ vs $\text{gemma}_{0.15}$ | 24.0 | 0.00000 |

## 7 Discussion and Conclusion

In this work, we introduced background temperature ($T_{\text{bg}}$) as a quantitative lens on implementation-dependent variability in Large Language Models. Even under nominally deterministic decoding, different inference environments induce statistically measurable randomness in the generated outputs. Our contribution is twofold: (i) we formalized $T_{\text{bg}}$ as an equivalent temperature that best matches observed variability against a controlled reference, and (ii) we proposed and empirically demonstrated a practical protocol for estimating it through variability metrics computed over repeated generations. Across multiple models, providers, and configured temperatures, our pilot studies show that $T_{\text{bg}}$ captures systematic and interpretable differences. First, background temperature varies substantially across model–provider combinations, rather than merely across model architectures. For example, the same model deployed via different cloud providers can differ by an order of magnitude, suggesting that infrastructure plays a central role in inference stability. Second, $T_{\text{bg}}$ exhibits a monotonic relationship with the sampling temperature $T$, indicating that it reflects effective output stochasticity beyond the configured temperature parameter. Third, temporal experiments show that $T_{\text{bg}}$ is sensitive to time-dependent factors, reinforcing that inference environment includes load patterns and low-level system behavior, not only model and API configuration. Finally, systems with similar estimated background temperature produce statistically similar distributions of output variability on unseen prompts, suggesting that $T_{\text{bg}}$ can serve as a stable proxy for characterizing response variability. Thus, we can conclude that using background temperature, rather than relying on ad hoc observations of nondeterministic outputs, provides a scalar quantity that practitioners can report, monitor, and compare across deployments. It is essential to emphasize that background temperature must not be interpreted as the model's actual sampling temperature. Rather, it is a single scalar that summarizes the net effect of implementation-induced noise by matching the variability statistics of the tested system to those of controlled reference runs. While alternative metrics such as perplexity, diversity scores, or edit-distance statistics each capture only a fragment of model variability, background temperature offers a unified and interpretable measure that aggregates all sources of randomness—hardware behavior, software kernels, concurrency effects, and numerical precision—into one coherent quantity. The methodology introduced here also comes with important limitations. A primary challenge is the lack of a truly ideal reference configuration: current estimates require comparisons against quasi-ideal, stable setups, which inevitably introduce model-dependent effects. While averaging over multiple reference systems helps reduce this influence, future work should focus on identifying well-defined, highly stable reference models and documenting their characteristics. Scaling the approach is another open issue. Although we employed multiple variability metrics, our experiments rely on relatively small prompt sets, and broader, domain-diverse prompt collections will be required for more robust estimates. In addition, background temperature values depend on several design choices, including the distance metric, the temperature grid, and prompt selection. Establishing shared conventions for these components — particularly through a standardized set of stable reference models and common evaluation settings — would allow background temperature to become a practical and genuinely comparable method for quantifying output variability in LLM systems.

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
