# OpenReview forum: "Introducing Background Temperature to Characterise Hidden Randomness in Large Language Models"
_TMLR — Accepted by TMLR_

### Review · Reviewer_1Q9j · 2025-10-31

**Summary Of Contributions:**

DISCLAIMER: This reviewer is not an expert on LLMs.

**Summary**
This paper proposes to quantify the non-determinism of LLM even at sampling temperature $T=0$ via the notion of background temperature. This is supposed to be the temperature at which the outputs of an ideal LLM, deterministic at $T=0$, would match the outputs of the LLM in question. The paper further proposes  a pipeline for estimating this background temperature and runs pilot experiments.

**Strengths**
- S1: The phenomenon of non-determinism is interesting and very relevant to trust and safety concerns around LLMs.
- S2: Aiming to quantify this non-determinism is interesting and the proposed method of computing a background temperature seems novel.


**Weaknesses**


This paper seems to be in a *very* preliminary state. Experiments are not thoroughly conducted, the presentation is suboptimal , and the paper structure is very unusual to say the least.

*Experiments:*
In Section 6 the authors give a blueprint of a possible pipeline for computing background temperature. But the supporting experiments are very minimal and fall short of many desiderata in section 6. Only 2 reference models are used, only one variability metric, exact-match rate, only 200 non-diverse prompts (all from truthful_qa) are used. Overall, 4 LLM models are investigated, but their findings do not get compared or discussed (e.g. the claude model was even perfectly deterministic). Moreover, the few conducted experiments are not even consistent: The ChatGPT experiment used 200 prompts, but for gemini, grok and claude only 30 prompts were used.

Before comparing different remote and reference models, it would be good to first verify the plausibility of the proposed pipeline. For instance, one could take a model, e.g. llama, that can be run deterministically for $T=0$, and use it both as reference and remote model (in the remote setting with some temperature $T>0$). This way one could check if the number of prompts, the variability metrics, the number of responses etc are suitable to reliably recover the true temperature under which the remote model was run. If successful, one could take this setup one step further and put the remote variant in a non-deterministic environment and check how the estimation of the true temperature changes (both when remote is run at $T=0$ and $T>0$). Such baseline experiments seem important to then trust background temperate estimations across different model types.

In sum, the experiments do not support the pipeline suggestions and are also too limited for meaningful scientific contribution.


*Conceptual issues:*
Why can we model the temperature of one LLM by that of another / a set of other reference LLMs? Why is this even meaningful? Are even the vocabularies the same?

What is the upshot of using background temperature (with all the non-canonical choices and references involved) when one could just compute perplexity or other diversity measures of the outputs of the remote LLMs at $T=0$?

*Presentation:*
The paper uses lots of notation that is often hard to follow and sometimes not explained. For instance, how are $\hat{z}$ in Eq (3) defined? What is  $n$ in eq (7)? Why is $\hat{z}$ indexed by the generation step $i$ and not the token index $t$ in Eq (3)? Why is the deterministic decoding in eq (2) equivalent to non-deterministic decoding with other logits $\hat{z}$ eq (3)?

Moreover, some parts of the proposed pipeline remain rather vague. E.g. how should one engineer prompts so as to elicit near-ties among the top-k tokens?

*Other unusual aspects*:
The entire paper only has 10 references. The introduction is extremely short, shorter even than the abstract. It does not contain any motivation for the relevance of the work, no examples, and only minimal contextualization. Sections 4,5 are (like the introduction) only about 10 lines long (or a quarter of a page), but sections 6 and 7 are 3 and 5 pages long, respectively. This seems very unbalanced. The figures are huge compared to their information content. There are several odd stylistic choices, such as starting a sentence with "In fact, even if, in general,"

**Audience:**

No

**Audience Explanation:**

The general questions is very interesting and the proposed approach might have some merit. But in its current very preliminary form, I doubt that much can be gained from this paper, unfortunately.

**Broader Impact Concerns:**

I do not think a Broader Impact Statement is necessary. While non-determinism in LLMs is of broad impact, this paper tries to quantify it, which rather addresses concerns than raising new ones.

**Claims And Evidence:**

No

**Claims Explanation:**

The notion of background temperature remains vague, since the proposed pipeline for computing it contains several possible choices and vague descriptions. The experiments are minimal and insufficient to corroborate the proposed pipeline.

**Requested Changes:**

See weaknesses. In particular:

- Run thorough experiments that come close to the intended pipeline.

- Verify the pipeline in settings where true temperatures are available.

- Clarify the intended pipeline and its notation.

- Compare to simpler diversity measurements.

- Fix basic formatting aspects of the paper, such as section lengths, figure sizes, number of references.

---

> ### Author Response · Authors · 2025-12-11
>
> We thank the reviewer for the detailed feedback. Based on this feedback and the other reviews, and building on experiments we were already conducting, we have uploaded a substantially improved version of the paper, incorporating new experiments on locally installed models at different temperatures, inclusion of additional variability metrics besides exact-match fraction, reporting results for gpt-4.1-nano on both full and reduced prompt sets, clarification that $T_{bg}$ is a comparative measure rather than an intrinsic temperature parameter, improved formalization of the notation and mathematical expressions, and adjustments to figure sizes and textual explanations for clarity.
> - On limited experiments and pipeline verification: We have expanded the experimental setup to include two additional variability metrics and one additional reference model. We also verified the pipeline on locally run models where true sampling temperatures are known (Sections 6.2). This demonstrates that $T_{bg}$ captures relative variability reliably.
> - On modeling one LLM via another/reference model: $T_{bg}$ should not be interpreted as estimating the actual temperature of a model. Instead, it quantifies relative variability, allowing comparison across systems tested under the same conditions. Comparing variability statistics rather than token distributions avoids vocabulary mismatch issues, see Sections 6 and 7.
> - On meaning relative to simpler diversity metrics: While other metrics like perplexity or edit distance are valid, $T_{bg}$ provides a unified measure, encapsulating systemic noise from all sources (hardware, software) on a single, interpretable scale. This unification aids reproducibility and deployment control. We also added a comment about that in Section 7.
> - On presentation, notation, and structure: We substantively improved the formalization of Sections, clarified all notation, adjusted figure sizes, and added references and context where useful, while keeping the current structure to focus on clarity rather than length balancing.

---

> > ### Comment · Reviewer_1Q9j · 2025-12-21
> >
> > Many thanks to the authors for significantly overhauling their paper. Some sections, such as the parts of the theoretical setup are much clearer now. However, many of my concerns remain, unfortunately.
> >
> > 1. Parts of the notation still unclear: What are the exact functional forms of $F_T$ and $\epsilon_I$? Or are these only meant as vague placeholders?
> > 2. Experimental setup: Why were prompts used a different times for test and reference systems? Why are there no uncertainties reported on the main experiments in Table 1? Are the different background temperatures observed for DeepSeek on different providers significant?
> > 3. On-premise experiments: Thanks for validating the approach with an on-premise experiment. There still seems to be a large mismatch between the true temperature and the estimated background temperature. This may be due to a mismatch between test and reference models. I appreciate that background temperatures should not be confused with actual temperatures, but it would be reassuring that if the reference model and the test model were the same, the proposed procedure could reliably recover the true temperature used. Otherwise, prompt set sizes, choice of variability measures may be unsuitable.
> > 4. Prompt-set size dependence: On gpt-4.1.-nano, changing the prompt set size led to a seemingly large (but without uncertainties significance is hard to judge) change in background temperature (Table 1). Does this not point towards a too small prompt set size? Should one not use a prompt set large enough that estimates are stable?
> > 5. Conceptual issues: The authors claim that background temperature is a "unified and interpretable measure". But since the authors also caution against relying on exact values of the background temperature or even viewing it as some sort of actual temperature, I am not sure how to interpret it exactly. I also do not get why it encompasses more randomness than alternative measures.
> >
> > Overall, I still think that the authors address a highly relevant topic. But I remain unconvinced by their proposed approach both due to conceptual issues and issues in the experimentation. Therefore, I cannot recommend acceptance, unfortunately.

---

> > > ### Author Response · Authors · 2025-12-29
> > >
> > > Thank you for the detailed follow-up. Below are our answers to your comments.
> > > 1.  **Notation**: $F_T$ and $\epsilon_I$ are intentionally abstract, denoting respectively the temperature-induced transformation of a probability distribution and the net effect of implementation-level perturbations induced by the inference environment. An explicit functional form for these operators is not known in practice, because it depends on the specific model (often proprietary) and, in the case of $\epsilon_I$, on inaccessible sources of nondeterminism. The goal is therefore not to specify them explicitly, but to characterize their observable effect via an equivalent temperature.
> > > 2.	**Experimental setup and uncertainties**: The number of samples per prompt differs between systems under test (100) and reference models (32). This asymmetry is intentional. Reference models are run locally in a highly stable inference environment, so increasing the number of samples does not significantly change their variability statistics, which are robust to unequal sample counts. Moreover, reference runs require sweeping multiple temperatures and models, and are therefore computationally expensive. The values reported in Table~1 are point estimates of $\bar{T}_{bg}^0(s)$, obtained by averaging over prompts, reference models, and variability measures. They should not be interpreted as exact system constants. Uncertainty arises primarily from prompt selection and environmental variability rather than from sampling noise in the reference runs. Estimating uncertainty reliably would require repeating the analysis across multiple prompt sets, reference models, and variability measures, which we leave for future work.
> > > 3.	**On-premise validation**: Background temperature is not intended to recover the configured sampling temperature. The two quantities have fundamentally different roles: sampling temperature is an implementation-level parameter controlling logit transformation, whose effect is model- and implementation-dependent, whereas background temperature is an output-level quantity summarizing the observed variability of final responses. As a result, identical sampling temperatures can induce very different variability across models, while systems with similar output variability may correspond to different sampling temperatures. What is crucial for validation is not equality but monotonicity: across all tested systems, background temperature increases monotonically with the configured sampling temperature, showing that it consistently tracks effective response variability while remaining model-agnostic.
> > > 4.	**Prompt-set size**:We agree that larger prompt sets yield more stable estimates. The comparison between $\Pi^{30}$ and $\Pi^{200}$ was included precisely to illustrate estimator sensitivity. This is a limitation of the current pilot study, not a claim that small prompt sets are sufficient in general. To make background temperature a reliable parameter for comparing variability across different model implementations, larger and more diverse prompt sets should be used, spanning different prompt types and generation regimes.
> > > 5.	**Conceptual issues**: The background temperature should be interpreted as an equivalent temperature. It does not represent a literal sampling parameter, but rather maps implementation-level noise onto a familiar and intuitive scale—the temperature of an idealized reference system whose output variability best matches the observed data. It is a "unified" measure because it aggregates various fragmented metrics (such as exact-match rates, edit distances,...) into a single scalar. It does not encompass "more" randomness than alternative measures; rather, it summarizes the net effect of all sources of nondeterminism—including hardware behavior, software kernels, and batching effects—into one coherent quantity. This allows practitioners to benchmark and compare stability across different models, providers, and hardware environments using a consistent and interpretable scale.

---

### Review · Reviewer_YqG6 · 2025-11-03

**Summary Of Contributions:**

This paper introduces the concept of "background temperature" ($T_{bg}$) to formalize and quantify the nondeterminism observed in Large Language Models (LLMs) even at a nominal decoding temperature of $T=0$. The authors propose that implementation-level factors (e.g., floating-point arithmetic, kernel choices) induce a stochastic perturbation on output probabilities, which is equivalent to decoding from an ideal, deterministic model at some non-zero temperature. The key contributions are the formal definition of this background temperature, a detailed empirical protocol for its estimation using a reference model, and a set of pilot experiments applying this protocol to several proprietary LLMs.

**Audience:**

Yes

**Audience Explanation:**

The problem of nondeterminism in LLMs, even in supposedly deterministic settings, is a significant practical issue for reproducibility, evaluation, and deployment in high-stakes applications. Practitioners and researchers in the TMLR audience are acutely aware of this challenge. The paper's proposal to frame this issue in terms of a "background temperature" is novel, intuitive, and provides a potentially powerful conceptual tool for reasoning about and measuring this hidden randomness. While the current execution has some notable limitations, the core idea and the problem it addresses are highly relevant and would certainly be of interest to the community. The extensive discussion of factors causing nondeterminism and the outline of a measurement protocol, even with its current issues, will spark valuable discussion and could inspire more rigorous future work.

**Broader Impact Concerns:**

The work focuses on the technical problem of reproducibility and does not appear to have direct negative broader impact concerns. If this method were to be used to "certify" models as having low background temperature, it would be critical that the measurement protocol be sound and standardized, as a flawed protocol could be used to mislead users about the reliability of a model. However, this is an issue of correct application rather than an inherent ethical problem with the research itself. The paper is transparent in its methods, and no Broader Impact Statement seems necessary at this stage.

**Claims And Evidence:**

No

**Claims Explanation:**

The central claims of the paper are not supported by sufficiently convincing evidence due to a combination of significant methodological limitations, a lack of theoretical justification, an insufficient scale of experimentation, and issues with the presentation of the evidence.

First, the empirical protocol makes a key methodological choice that, while pragmatic, significantly complicates the interpretation of the results. The authors propose to estimate the background temperature of a target model (System B, e.g., `gpt-4.1-nano`) by comparing its output variability at $T=0$ to the variability of a completely different reference model (System A, e.g., `SmolLM3-3B`). The authors acknowledge the difficulty of obtaining a "quasi-ideal" reference of the same model. However, the chosen alternative of using a different model introduces a significant confound. This makes it difficult to isolate the implementation-dependent nondeterminism of System B, as the final measurement reflects a combined effect of: (1) the actual system noise of B, (2) the intrinsic differences between the models' output distributions, and (3) the differing sensitivities of each model's architecture to temperature scaling. The authors' own results illustrate this challenge: the estimated $T_{bg}$ for `gpt-4.1-nano` changes from 0.05 to 0.10 simply by switching the reference model. The proposed solution of averaging these results is a heuristic that highlights the sensitivity of the measurement to the reference choice, rather than resolving the ambiguity. This makes it challenging to interpret the final number as an intrinsic property of the system under test.

Second, the paper lacks theoretical grounding for its central claims. The core modeling assumption, stated in Equation 7 ($F_I^0(P) \approx \epsilon_I(P) \approx F_{T_n(I)}(P)$), posits that implementation-dependent noise acts *as if* it were a temperature transformation on the logits. This is a strong claim presented as an axiom without any theoretical proof or derivation. It is entirely possible that system-level perturbations induce changes to the probability distribution that are not well-approximated by a simple temperature scaling. The claim in Section 6.5 that the estimator $\hat{T}_bg$ will converge to the true $T_{bg}$ is also stated without proof, leaving it as an assertion.

Third, the pilot experiments are somewhat small in scale and narrow in scope to support the paper's broad conclusions. The primary experiment uses 200 prompts, while subsequent experiments use only 30. Such sample sizes may not be sufficient to yield robust estimates. Moreover, the evaluation relies exclusively on a single, coarse metric: the exact-match fraction. This metric is useful but cannot capture more subtle distributional shifts. The paper proposes a richer set of metrics in Section 6.4 but fails to employ them, missing an opportunity to provide more convincing evidence.

Finally, the presentation of the evidence could be improved. The paper's layout, particularly with pages 8 and 9 consisting entirely of figures, disrupts the narrative flow. While visual aids are crucial, integrating them more closely with the relevant text would improve readability. This current layout separates the results from their description and could suggest that the supporting textual analysis is less substantial than the visual presentation.

**Requested Changes:**

**Critical Changes:**

1.  The experimental methodology should be revised to address the significant confound introduced by using a different model as the reference. The current approach makes it difficult to isolate the variable of interest. A critical revision would be to re-run experiments using a protocol that compares the system under test to a "quasi-ideal" configuration of the *same* model, as described in the first paragraph of Section 6. If this is not feasible for remote APIs, the paper must more thoroughly discuss this limitation and be much more cautious in its claims, framing the results as a model-pair-dependent metric rather than an intrinsic property of the test model.
2.  The mathematical formalism needs to be tightened. The definitions in Sections 4 and 5 rely on imprecise "$\approx$" symbols (e.g., Equation 7). The concept of an "equivalent temperature" should be defined more rigorously, for example, by formally stating that it is the temperature that minimizes a specific divergence metric, which connects the theory more directly to the empirical protocol.
3.  The definition of background temperature in Equation 8, $T_{bg} := \mathbb{E}_{I \in \mathcal{I}}[T_n(I)]$, lacks rigor. The expectation is taken over a set $\mathcal{I}$ without defining a probability measure on it. Please specify the distribution over which the expectation is taken or clarify how this expectation should be interpreted.

**Strengthening Changes:**

1.  The pilot experiments are limited to a single metric (exact-match fraction). The paper would be much stronger if the experiments were expanded to include some of the other metrics proposed in Section 6.4 (e.g., distributional divergence, edit distance). This would help assess whether the $T_{bg}$ estimate is robust to the choice of metric.
2.  The layout of the paper should be improved to better integrate figures with the text. Having two full pages (8 and 9) dedicated only to figures disrupts the reading experience and separates the evidence from its interpretation.
3.  The finding that `claude-sonnet-4` has an estimated $T_{bg}=0$ is very interesting and warrants more discussion. Does this truly imply it is deterministic, or is it a limitation of the measurement protocol or the specific prompts and metric used? Exploring this further would add significant value.

---

> ### Author Response · Authors · 2025-12-11
>
> We thank the reviewer for the detailed and constructive feedback. Based on this feedback and the other reviews, and building on experiments we were already conducting, we have uploaded a substantially improved version of the paper, incorporating new experiments on locally installed models at different temperatures to assess variability, inclusion of additional metrics beyond exact-match fraction, a more rigorous mathematical formalization of background and equivalent temperature, clarification that $T_{bg}$ depends on the chosen reference models rather than being an intrinsic property, and improved figure integration and textual explanations reflecting these updates.
> - On the dependence of $T_{bg}$ on the reference model: We clarified that the estimated background temperature is a function of the chosen reference models. Without a dedicated quasi-ideal protocol, $T_{bg}$ should be interpreted as a comparative measure of response stability across systems rather than an intrinsic property of a single model.
> - On quasi-ideal comparisons While running a quasi-ideal configuration is not feasible for proprietary models, we conducted new experiments on local models at different temperatures, showing that $T_{bg}$ increases accordingly with temperature, even if absolute values do not exactly match (Section 6.2).
> - On metrics and pilot experiments: The new experiments include two additional metrics besides exact-match fraction and one more reference model, increasing robustness and interpretability of the results.
> - On mathematical formalism: The new version (Section 4) provides a fully rigorous formulation of $T_{bg}$ and equivalent temperature, specifying expectations and probability distributions explicitly.
> - On figure layout: We removed some figures to better integrate experimental results with text.
> - On discussion of the case claude-sonnet-4: Regarding claude-sonnet-4, it is deterministic on the tested prompt set, suggesting that the AWS-claude-sonnet-4 pair allows purely greedy decoding at temperature 0, though this may not generalize to all prompts or token limits.

---

### Review · Reviewer_9HtF · 2025-12-02

**Summary Of Contributions:**

To quantify intrinsic variability in LLM outputs — even under greedy decoding, where responses are expected to be deterministic — the authors introduce the notion of background temperature  Tbg and propose proxies for estimating it using a reference LLM. Although identifying a stable and reliable reference model poses challenges, I believe their analysis represents a meaningful step toward establishing comparable metrics for LLM certification.

**Audience:**

Yes

**Audience Explanation:**

LLM certification is a deep and impactful area where the presented results can be of interest.

**Broader Impact Concerns:**

I do not have any specific concern in this line.

**Claims And Evidence:**

Yes

**Claims Explanation:**

The metrics and experimental setups are generally well presented. The authors’ main contribution is the introduction of the background temperature, which is evaluated across several use cases. While I would have appreciated deeper analysis or more substantive conclusions from the comparisons, it is fair to say that the authors deliver on what they claim: they introduce the metric and report results accordingly.

**Requested Changes:**

- Out of the comparison of the metric in Table 1, can we draw certain conclusions about how the LLM architecture/training influences the background temperature?
- Can the authors elaborate in more detail why the batch size influences LLM output distribution in model validation/test? This is referered to a source of variability in LLM outputs but I don't see how it may influence. Similarly, I miss some introduction to batch-invariant kernels. I believe the paper would really improve if a new section summarizing sources of variability in LLMs is introduced, detailing all these aspects.
- Please elaborate some conclusions out of the comparison in Table 1. How the LLM structure/architecture/training may influence the background temperature.

---

> ### Author Response · Authors · 2025-12-11
>
> We thank the reviewer for the careful reading and constructive comments. Based on this feedback and the other reviews, and building on experiments we were already conducting, we have uploaded a substantially improved version of the paper, incorporating several new experiments that reinforce the robustness of $T_{bg}$, a reference model to improve its estimation, additional measures of variability, analyses on estimation timing, a more rigorous mathematical formulation, and updated introduction and conclusions reflecting the expanded results.
>
> - On LLM structure/architecture/training: Additional experiments show that $T_{bg}$ is mainly influenced by the inference environment rather than model architecture or training; for example, the same model deployed on different providers yields different $T_{bg}$ estimates (Section 6.1).
> - On batch size, batch-invariant kernels, and sources of variability: We now summarize sources of variability in LLM outputs (Section 2.1), explain how batch size can affect outputs through resource interactions, and briefly discuss batch-invariant kernels to mitigate such effects (Section 2.2).

---

> > ### Comment · Reviewer_9HtF · 2026-01-07
> >
> > I want to thank the authors for further experimental results and associated analysis. I believe this contribution represents an interesting step towards LLM certification and evaluation on its inherent variability.

---

### Decision · Action_Editor_A65y · 2026-01-26

**Recommendation:** Accept as is

**Audience:**

Yes

**Audience Explanation:**

Yes, this is a complementary question to the normal definition of temperature which influences how we view/analyze our systems' outputs.

**Claims And Evidence:**

Yes

**Claims Explanation:**

Two reviewers (post-revision) agree that these are now supported.  The work was very preliminary when submitted ask as such lacked necessary details, definitions, experiments, etc.  Broadly, these have been addressed but the work should still be viewed as preliminary but relevant, pilot studies cannot be shown to generalize as broadly but still point in a useful direction.

---

> ### Author Response · Authors · 2026-02-13
> **camera-ready**
>
> Hi,
> thank you for the great news. We have uploaded the camera-ready version of the paper.
> Please let us know if you need anything else from our side.
> Best regards,
> The authors